# Comparison of Cholangiocarcinoma and Hepatocellular Carcinoma Incidence Trends from 1993 to 2012 in Lampang, Thailand

**DOI:** 10.3390/ijerph19159551

**Published:** 2022-08-03

**Authors:** Pianpian Cao, Laura S. Rozek, Donsuk Pongnikorn, Hutcha Sriplung, Rafael Meza

**Affiliations:** 1Department of Epidemiology, University of Michigan, Ann Arbor, MI 48109, USA; rmeza@umich.edu; 2Department of Environment Health Sciences, University of Michigan, Ann Arbor, MI 48109, USA; rozekl@umich.edu; 3Department of Otolaryngology, University of Michigan, Ann Arbor, MI 48109, USA; 4Lampang Cancer Hospital, Lampang 52000, Thailand; donsukp@hotmail.com; 5Department of Epidemiology, Faculty of Medicine, Prince of Songkla University, Songkhla 90110, Thailand

**Keywords:** liver cancer, cholangiocarcinoma, hepatocellular carcinoma, *O. viverrini* infection, liver fluke, Jointpoint analysis, age-period-cohort model

## Abstract

Liver cancer is the most common cancer in Northern Thailand, mainly due to the dietary preference for raw fish, which can lead to infection by the parasite, *O. viverrini*, a causal agent of cholangiocarcinoma. We conducted a temporal trend analysis of cross-sectional incidence rates of liver cancer in Lampang, Northern Thailand. Liver cancer data from 1993–2012 were extracted from Lampang Cancer Registry. The multiple imputation by chained equations method was used to impute missing histology data. Imputed data were analyzed using Joinpoint and age-period-cohort (APC) models to characterize the incidence rates by gender, region, and histology, considering hepatocellular carcinoma (HCC) and cholangiocarcinoma (CCA). We observed a significant annual increase in CCA incidence and a considerable decrease in HCC incidence for both genders in Lampang. The APC analysis suggested that CCA incidence rates were higher in older ages, younger cohorts, and later years of diagnosis. In contrast, HCC incidence rates were higher in older generations and earlier years of diagnosis. Further studies of potential risk factors of CCA are needed to better understand and address the increasing burden of CCA in Lampang. Our findings may help to draw public attention to cholangiocarcinoma prevention and control in Northern Thailand.

## 1. Introduction

Liver cancer, which includes hepatocellular carcinoma (HCC), cholangiocarcinoma (CCA), and other histology types, is the most common cancer among males and the third most common cancer among females in Thailand [1,2,3,4]. Liver cancer incidence varies widely by region and is substantially higher in the North, especially in the Northeast [1,2,3,4]. The histological distribution of liver cancer also greatly differs from region to region, with CCA being the most common type of liver cancer in Northern Thailand, whereas HCC is the dominant type in other areas [1,2,3,4].

The high prevalence and incidence rates of CCA in Northern Thailand are closely related to an endemic parasitic disease, Ophisthorchis viverrini infection. *O. viverrini* colonization in the liver increases the risk of CCA by causing chronic bile duct inflammation, wounding the biliary epithelia, and excreting parasitic proteins that promote cell proliferation [5]. *O. viverrini* infection prevalence varied widely by region in Thailand, being 34.6%, 6.34%, 5.59%, and 0.01% in the Northeast, Central, North, and South, respectively, according to a nationwide survey in 1981 [6]. This geographical difference in the infection prevalence comes from the distinct dietary preference for a traditional Thai dish made from raw or undercooked cyprinoid fish, which may be contaminated with *O. viverrini* larvae. This dish is very popular in the Northeast, while people rarely consume it in the South [7,8].

Nearly all national *O. viverrini* infection control efforts were concentrated in the Northeast, especially the Khon Kaen province, where raw fish consumption is the most popular; these control programs successfully reduced infection prevalence and, consequently, CCA incidence [1,2,3,4,9]. However, people in the North still consume raw fish dishes when there are big family gatherings or holidays [7], leading to potentially high risks of *O. viverrini* infection and thus cholangiocarcinoma.

As reported by Cancer in Thailand, the liver cancer incidence rate in the North is three times higher than that in the South; Lampang, a Northern province, currently has the highest liver cancer incidence among provinces in the North [2]. However, few *O. viverrini* interventions have been conducted in Lampang. Hence, it is necessary to investigate the trends of liver cancer and *O. viverrini* prevalence in Lampang to assess the need for running similar infection control programs as in the Northeast. The goal of this study was to analyze data from the Lampang cancer registry using Joinpoint regression and age-period-cohort models to characterize temporal trends of liver cancer incidence by histology, sex, age, period, and cohort.

## 2. Materials and Methods

### 2.1. Data Sources

Data for this study were obtained from the Lampang cancer registry (LCR), Thailand. Not to miss the cases diagnosed by radiography as cholangiocarcinoma defined by the Bismuth–Corlette classification [10] to be intra- and extrahepatic subgroups, we identified all primary liver and biliary cancer cases from the LCR based on ICD-O-3 codes. All cases were further classified into four histology groups using ICD-O-3 topography and morphology codes—hepatocellular carcinoma (HCC; C22.0, 8170:8175), cholangiocarcinoma (CCA; C22.1, C24.0, C24.8 or C24.9), other (OTH; C22.0 with morphology codes other than 8000:8005 and 8170:8175, or C24.1) and unknown (UNK; C22.0, 8000:8005) [11,12]. Numbers of liver cancer cases by histology and sex are presented in Table 1. Descriptive characteristics of available variables in the dataset can be found in Table A1. This study is a secondary analysis of de-identified cancer registry data; thus, the institutional review board approval is not required. 

### 2.2. Multiple Imputation

The percentage of liver cancers with unknown histology is very high in both males (53%) and females (46%). We thus conducted multiple imputation to further classify the unknowns into HCC, CCA, or OTH using a multinomial logistic regression model with all available covariates in the cancer registry: sex, age, year at diagnosis, and living area [12,13]. Even though in practice, three to five imputed datasets are sufficient to obtain stable results, experts recommend that for datasets containing a large percentage of missingness, more than 40 imputations are needed to obtain a less than 1% power fall-off [14]. We then generated 200 imputed datasets to get more accurate estimates for the number of cancer cases for each histology, sex, age, and year of diagnosis combination. We used the average number of cases from the 200 imputed datasets for subsequent analyses. We further demonstrated how the percentages of HCC by sex in 1993 varied according to the number of total imputation (Figure A1). Multiple imputation was conducted using the ‘mice’ package version 3.12 [15] in the R statistical software version 3.6.3 (R Foundation for Statistical Computing, Vienna, Austria) [16].

### 2.3. Joinpoint Analysis

To analyze secular trends of liver cancer incidence in Lampang by sex and histology, we performed a Joinpoint regression analysis on the age-adjusted incidence rates (AARs) from 1993 to 2012. We calculated age-specific incidence rates (ASRs) for eighteen 5-year age groups (0–4, 5–9, 10–14, …, 80–84, 85+) and used the world population (Segi 1960) as the standard adjustment population. Joinpoint regression analysis enables us to find any time point (joinpoint) where a significant change in trend occurs and to quantify the trends between joinpoints. We set the maximum number of joinpoints to be 5 with a minimum of 0. A log-linear model was used to fit the data as it allows us to obtain the average annual percent change (AAPC) between joinpoints. Parallel pairwise comparisons were conducted to assess if the AAPCs for a histology and gender combination were the same before and after imputation. Joinpoint analysis was performed using the Joinpoint Trend Analysis Software version 4.9.1.0 (National Cancer Institute, Bethesda, MD, USA) [17].

### 2.4. Age-Period-Cohort Analysis

Age-period-cohort (APC) models were used to study how age, calendar year (period), and birth-cohort correlate with liver cancer age-specific incidence risk. The APC model assumes a log-linear relationship between the incidence rate and age, period, and cohort, where incidence is assumed to follow a Poisson distribution. We conducted the analysis by sex and histology and restricted it to ages between 35 and 84. Age-specific incidence was then calculated for 50 single-year age groups and 20 calendar years (1993, 1994, …, 2012). This results in 69 single-year birth cohorts from 1909 to 1977. Age at diagnosis (A), year of diagnosis (P) and birth year (C) are linearly correlated (C = P-A), causing the well-known non-identifiability problem [18]. To circumvent it, we first fitted two-effect models (age-period and age-cohort) to the data and then fitted the remaining effect (cohort and period) to the residuals from the previous models. We call the model fitting age and period first as AP-C model and the other as AC-P model. APC models were run only for HCC and CCA imputed data. Age, period, and cohort effects were fitted with natural splines, with 9 degrees of freedom for age, 4 for period and 7 for cohort. APC model analysis was conducted using the ‘Epi’ package (Version 2.44) in R [19].

## 3. Results

### 3.1. Multiple Imputation

Before imputation, 36% of liver cancers were classified as CCA, 9% as HCC, 3% as other histology types, and 53% as unknown among males; the percentages of CCA, HCC, OTH, and unknown among females were 44%, 5%, 5%, and 46% respectively (Table 1). The percentages of the three histology types by gender after imputation are also summarized in Table 1. We observed that the imputation procedure roughly brought all the unknowns into the three other types proportionally to their original sample sizes, with CCA being the most common type for both genders before and after imputation. In addition, Figure A1 shows the average proportion of HCC in 1993 by sex from the imputed datasets across the number of imputations, which suggests that the proportions stabilize after about 100 imputations.

### 3.2. Jointpoint Analyses

Joinpoint analyses found no significant point of change in AAR trends for all sex and histology combinations before and after imputation (Figure 1; Table 2), indicating a monotonic trend. For both sexes before and after imputation, there was a significant annual increase of AARs for CCA incidence (5.0 AAPC in males and 2.02 in females after imputation) and, in contrast, a significant yearly decrease of AARs for HCC incidence (−7.30 AAPC in males and −10.29 in females after imputation) from 1993 to 2012. The AARs of unknown histology type increased over the years among males but decreased significantly among females. For given histology, males had higher AARs than females.

According to the parallel pairwise comparisons, AAPCs are tested to be the same before and after imputation for all histology (HCC, CCA, or OTH) among males (Table A2). Assuming parallelism, the AARs of HCC among males decreased by 7.5% annually from 1993 to 2012, while the AAR of CCA among male increased by 5.6% annually from 1993 to 2012. However, with no reported HCC incidence cases in the year 2008 and 2009 and no reported OTH cases in 1997 and 1999, we could not fit a log-linear model to the data of HCC and OTH before imputation. Therefore, before-and-after imputation comparisons can be made only for CCA, and it was found to be not parallel, with the after-imputation AAPC estimate more conservative (Table 2 and Table A2).

### 3.3. Age-Period-Cohort Analyses

We restricted the APC analyses to imputed HCC and CCA data. The Akaike information criteria (AIC) and deviance values of age-cohort (AC), age-period (AP), and age-period-cohort (APC) models by sex and histology were presented in Table 3. APC models have the lowest deviance, indicating it fits the data the best. However, the APC model does not always have the lowest AIC value, which penalizes the number of parameters in the model. The AP model for HCC has the lowest AIC for males and females, while the AC model has the lower AIC than the AP model for male and female CCA. Taking both AIC and deviance into consideration, we presented the AP-C models for HCC and the AC-P models for CCA below.

We observed two peaks in HCC age-specific incidence rate for both males and females (Figure 2). The first peak occurred at around age 50 for both sexes, while the second peak occurred in the early 60s for females and in their late 60s for males. In terms of the period (calendar year), the male HCC relative incidence increased slightly until 1996 and then decreased continuously. For females, the relative incidence decreased sharply from 4.84 to 1 until 2005, when the rate of decline slowed down. The cohort effect hovered around a relative risk of 1. This confirms the conclusion drawn from Table 3 that AP model alone can explain the ASR pattern of HCC well, such that adding cohort into the model does not improve model fit significantly.

The age effects estimated by AC-P models of CCA had similar patterns for males and females: increasing consistently by age (Figure 3). The AC-P model for male CCA estimates continuously increasing incidence by birth cohort. For instance, the CCA incidence for males born in 1977 was six times higher than for those born in 1945. The relative incidence of female CCA also increased by birth cohort but at a lower rate than males. The period effects stay close to the relative risk of 1, consistent with the fact that AC models fit the data well without the need for period effects.

## 4. Discussion

Our study suggested that HCC incidence in Lampang decreased significantly since 1993, with a more dramatic decreasing trend in females (AAPC of −10.29%) than in males (AAPC of −7.3%). The temporal trend of HCC can be explained by age and period effects alone, suggesting that calendar year rather than the year of birth better correlates with HCC incidence. In contrast, CCA incidence has increased substantially over the past 20 years in males and females, with AAPCs of 5% and 2.02% for males and females respectively. Age-Period-Cohort models show considerable increases in CCA by birth cohort, with cohort effects correlating better with CCA trends than period effects.

The decreasing trend of HCC in Lampang identified in our study is consistent with that found in other provinces in Thailand [8,20]. The decrease might be related to the reduction in the prevalence of several established risk factors for HCC, including Hepatitis B virus (HBV), Hepatitis C virus (HCV) infections, and smoking [21]. A national HBV immunization program for all newborns was implemented in Thailand in 1992 [22]. This program has helped lower the HBV carrier rate among children and has been shown to reduce the HCC incidence among children [22,23]. However, the reduction of HCC incidence in children may help explain only a little of the significant decline in overall HCC incidence as HCC is rare in children [24]. We may speculate that, after the introduction of the universal immunization program, adults get some benefits due to herd immunity, which might further drive the prevalence of HBV in the whole population. However, limited data on the HBV infection rates among adults and related health outcomes are available in Thailand. Further studies are needed to test this hypothesis.

Reductions in HCV and HBV prevalence due to improved personal and medical hygiene might also account for the decreased HCC rates [25]. HCV seroprevalence in Thailand has been reduced because of more effective HCV treatment and improvements in clinical hygiene, such as using disposable syringes and strict medical equipment sterilization [25]. Furthermore, Thailand’s Center for Disease Control has put emphasis on promoting condom use. The “100% Condom Program” was launched in 1991 to relieve the burden of HIV and other sexually transmitted diseases [26]. This program might have led to further reductions in HBV and HCV prevalence since these viruses can be sexually transmitted [27].

Reduction in smoking prevalence might also play a role as smoking is a risk factor for HCC. Thailand is the first Asian country to implement strict tobacco control regulations, including bans on tobacco advertising and smoking in public in 1992 and increasing taxation on tobacco products since 1994, significantly reducing smoking rates in Thailand [28].

All these factors could work synergistically to reduce the HCC incidence in Lampang. Results from our AP-C models for HCC show that the incidence rate of HCC decreases by calendar year, and period effects alone can explain most of the declining patterns in HCC. This agrees with our speculation that all these universal programs that target the whole population in recent years have reduced the prevalence of risk factors for HCC, which has led to a decline in HCC incidence. However, it is worth noting that all these speculations are based on national data, which may not capture the specific context of the Lampang province.

Although sharing similar risk factors as HCC, such as HBV/HCV infections and smoking, the incidence of CCA is rising dramatically in Lampang. Newer generations in Lampang have a higher risk of CCA than previously; for example, the AC-P model suggests that men born in 1970 have a four times higher risk of CCA than those born in 1945. Younger generations may have undergone behavioral changes, leading to an elevated risk of CCA. For instance, with the popularization of western fast food in Thailand since the late 1970s [29], younger birth cohorts have been exposed to higher-fat and lower-fiber diets than older birth cohorts. A high-fat and low-fiber diet is known to be a risk factor for gallstones [30], which has been found to have a significant positive association with the risk of CCA [31]. Therefore, dietary changes by generations might be contributing to the increase in the incidence of CCA. Another likely explanation for the difference observed in CCA and HCC trends is that the prevalence of *O. viverrini* infection, a distinct and established risk factor for CCA, has increased for recent generations in Lampang. This could be due to limited awareness education campaigns and control programs to prevent *O. viverrini* infection in Lampang, Thailand, in contrast to the Northeast. Conversely, the CCA incidence in Khon Kaen has declined significantly since 2002, 20 years after the launch of *O. viverrini* infection control program in the province [32].

Our study has some limitations. There are a large fraction of cases with unknown histology in the data, consistent with data from other Asian countries [33]. Since early detection of liver cancer is still a challenge in these countries, patients are often diagnosed in advanced stages when only palliative care can be given, and thus, histology determination is usually not performed [21,32,33]. To address this limitation, we used multiple imputation techniques to re-classify cases with unknown histology as HCC, CCA, or OTH. This method assumes that the data is missing at random (MAR), that is, that the probability of missingness depends only on observed variables but not unobserved ones [34]. Although this is a strong assumption that couldn’t be corroborated, the prediction of histology using available data may still be valid as variability in histology patterns could still be captured by existing variables, such as year of diagnosis and sex. Under the MAR assumption, multiple imputation should lead to unbiased results. However, we cannot completely rule out the possibility that histology depends on other unobserved factors, i.e., not missing at random (NMAR) [34]. Such violation could lead to biased estimates of the number of cases in HCC, CCA, and OTH. This is the main limitation of our study, nonetheless the bias in estimations may still be minor even under the NMAR scenario with a high percentage of missing (e.g., 80%) [35]. Future studies testing the validity of the model to predict the histopathology of unknown liver cancer cases is thus required. That being said, we conducted analyses with imputed and non-imputed data, finding consistent trends. Second, our study did not evaluate the causal relationships between liver cancer and potential risk factors. Future studies are needed to study possible risk factors for liver cancer, especially CCA, in Lampang.

Our study has a few strengths and important implications. It allows us to assess the influence of age, cohort, and period on liver cancer trends by histology, in addition to serving as the basis for future research, including studying possible risk factors for HCC and CCA, evaluating the burden of CCA in Lampang, and assessing the potential impacts of public interventions to control CCA burden. Furthermore, given the increasing trend of CCA, a province-wide *O. viverrini* infection prevalence study and a survey regarding risk factors of CCA are needed to corroborate the potential role of *O. viverrini* infection in CCA etiology in Lampang. This would provide critical information to determine optimal control strategies against CCA in Lampang. For instance, if *O. viverrini* human infections are found to be common in Lampang, similar public health interventions as those conducted in Khon Kaen in the 1980s and 1900s would be needed to reduce the burden.

## 5. Conclusions

Temporal trends of liver cancer were estimated for different histologic groups in Lampang, Thailand. While hepatocellular carcinoma incidence decreased significantly recently, cholangiocarcinoma incidence increased dramatically. The differences between HCC and CCA incidence trends justify the importance of conducting research related to CCA etiology in Lampang to determine the best course of action.

## Figures and Tables

**Figure 1 ijerph-19-09551-f001:**
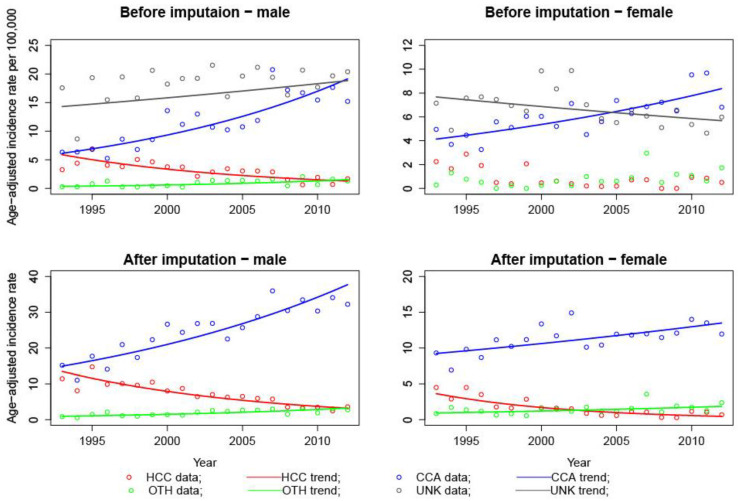
Age-adjusted incidence rate trends by sex and histology before and after imputation.

**Figure 2 ijerph-19-09551-f002:**
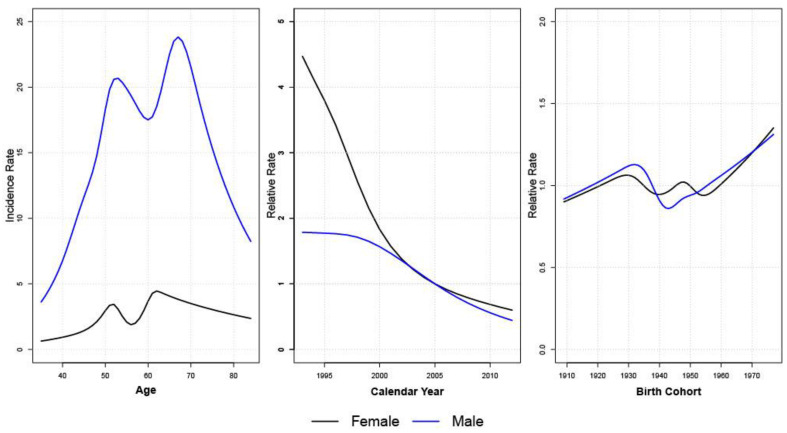
Age, period, and cohort effects of AP-C models for male and female HCC.

**Figure 3 ijerph-19-09551-f003:**
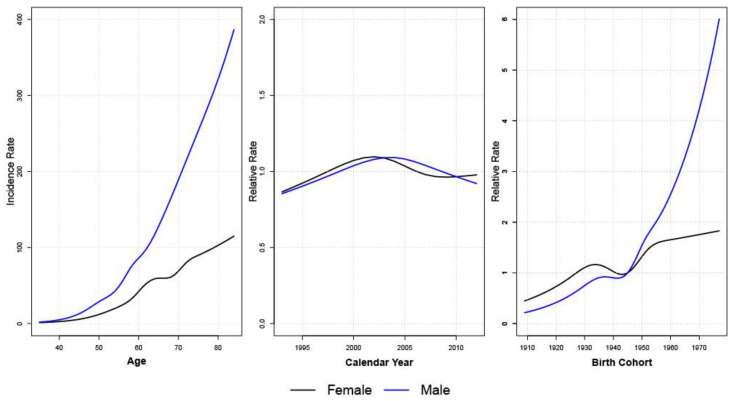
Age, period, and cohort effects of AC-P models for male and female CCA.

**Table 1 ijerph-19-09551-t001:** Number and percentage of liver cancer cases by sex and histology before and after ^1^ imputation.

Characteristics	Male Number (%)	Female Number (%)	Total Number (%)
Before Imputation			
HCC	294 (8.7)	76 (5.0)	370 (7.5)
CCA	1222 (36.1)	670 (43.7)	1892 (38.4)
OTH	89 (2.6)	83 (5.4)	172 (3.5)
UNK	1784 (52.6)	703 (45.9)	2487 (50.5)
After imputation			
HCC	609 (18.9)	144 (10.0)	753 (16.2)
CCA	2428 (75.5)	1159 (80.1)	3587 (77.0)
OTH	177 (5.5)	144 (10.0)	322 (6.9)

^1^ Average number of cases from 200 imputed datasets.

**Table 2 ijerph-19-09551-t002:** Average Annual Percent Change (AAPC) by sex and histology before and after imputation.

Gender	Histology	Year	AAPC (95% CI) Before Imputation	AAPC (95% CI) After Imputation
Male	HCC	1993–2012	−7.69 * (−10.6, −4.7)	−7.30 * (−8.8, −5.8)
	CCA	1993–2012	6.18 * (4.6, 7.8)	5.00 * (3.7, 6.3)
	OTH	1993–2012	8.19 * (2.9, 13.8)	6.64 * (3.9, 9.5)
	UNK	1993–2012	1.46 (−0.1, 3.0)	---
Female	HCC	1993–2012	N/A ^1^	−10.29 *
	CCA	1993–2012	3.78 * (2.3, 5.3)	2.02 * (0.9, 3.2)
	OTH	1993–2012	N/A ^1^	3.71 * (0.5, 7.0)
	UNK	1993–2012	−1.57 * (−3.1, −0.0)	---

* The AAPC is significantly different from zero at a significant level of 0.05. ^1^ Log-linear model cannot be fitted as data contains AARs equaling 0. CI = Confidence Interval.

**Table 3 ijerph-19-09551-t003:** Akaike information criteria (AIC) ^1^ and residual deviance for APC models.

Model	Female HCC	Female CCA	Male HCC	Male CCA
Akaike information criteria (AIC) for AC, AP, and APC models relative (difference) to the Age only model				
Age Cohort	−49.0	**−10.4**	−95.7	−151.7
Age Period	**−55.3 ^2^**	−6.8	**−105.5**	−143.8
Age Period Cohort	−41.8	−6.4	−100.3	**−154.7**
Residual deviance values for AC, AP, and APC models				
Age	372.0	981.9	656.9	1045.1
Age Cohort	309.0	957.5	547.2	879.4
Age Period	308.7	967.1	543.4	893.2
Age Period Cohort	**308.2**	**953.5**	**534.6**	**868.4**

^1^ AIC = −2 × log⁡ (likelihood) + 2 × number of estimated parameters, and a lower AIC value indicates a better-fit model. ^2^ Models with the lowest AIC or residual deviance values were bolded, indicating that the best-fit models.

## Data Availability

The datasets generated and/or analyzed during the current study are available from the corresponding authors on reasonable request.

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
