# Peer review of "Comparison of Cholangiocarcinoma and Hepatocellular Carcinoma Incidence Trends from 1993 to 2012 in Lampang, Thailand"

_ijerph, 2022, doi:10.3390/ijerph19159551_

Round 1
Reviewer 1 Report
Firstly, I would like to thank the authors for the opportunity of reading such an interesting paper. The retrospective cohort study titled ‘Comparison of cholangiocarcinoma and hepatocellular carci-2 noma incidence trends from 1993 to 2012 in Lampang, Thailand’, investigates current trends of liver cancer in Lampang, Northern Thailand.
The manuscript reported that HCC incidence in Lampang decreased significantly since 1993, with a more dramatic decreasing trend in females than in males, while CCA incidence has increased substantially over the past 20 years in males and females.
I suggest the authors consider the following comments to improve the quality of the manuscript. The following are the major and minor comments;
Major comments
- In the present study, we do not know whether the increased incidence of CCA in Lampang can be attributed to the increased infection rate of O.viverrini. In the abstract, the hypothesis and these results were confused. The authors should state only the facts of the present results.
Minor comments
Title and abstract
- Please clearly indicate the study design in commonly used terms.
Methods
- The author should list all confounders adjusted for in a multinomial logistic regression.
Results
- Please list details of each available covariate in the cancer registry in Appendix.
Author Response
Please see the attachment for a detailed response to each of the comments. Thank you!

Reviewer 2 Report
The authors analyzed the data from the Lampang cancer registry using multiple imputation by chained equations method, Joinpoint regression and age-period-cohort models to characterize temporal trends of liver cancer incidence by histology, sex, age, period, and cohort. They showed a significant increase in CCA and a considerable decrease in HCC. They suggest that the increase of CCA is due to an increase of O. viverrini infection.
The manuscript is interesting. I have only one comment.
Minor comments
#1. P4, line 136. “−7.69” may be mistake for “−7.30”.
Author Response
Thank you very much for the positive and detailed comment. We have revised the manuscript according to your suggestion.
